# A New Approach for Ocean Surface Wind Speed Retrieval Using Sentinel-1 Dual-Polarized Imagery

**Yuan Gao** [1,*], **Yunhua Wang** [1] and **Weili Wang** [2]

1 Faculty of Information Science and Engineering, Ocean University of China, Qingdao 266100, China; yunhuawang@ouc.edu.cn
2 Hainan Institute, Zhejiang University, Sanya 572025, China; wangweiliee@hotmail.com
* Correspondence: gaoyuan6926@ouc.edu.cn

**Abstract:** A synthetic aperture radar (SAR) has the capability to observe ocean surface winds with a high spatial resolution, even under extreme conditions. The purpose of this work was to develop a new method for wind speed retrieval with the combination of SAR dual-polarized signals. In this study, we collected 28 tropical cyclone imageries observed using the Sentinel-1 dual-polarization mode. These imageries were collocated with radiometer wind speed measurements and reanalysis of wind vector products. In the new method, the wind speed was set as the output. VV-polarized (vertical transmitting–vertical receiving polarized) normalized radar cross section (NRCS), incident angle, VH-polarized (vertical transmitting–horizontal receiving polarized) NRCS, and wind direction were set as the inputs. Based on different output combinations, wind retrieval models were developed with multiple linear regression (MLR). According to the validation and comparison, the proposed models performed better than the traditional piecewise VH-polarization geophysical model functions (GMFs). The impact of thermal noise on the retrieval of low wind speeds (<10 m/s) could be partially reduced. The input of wind direction is unnecessary if the combination of VV- and VH-polarized imageries has been utilized. These results suggest that the use of MLR and the dual-polarization combination can improve SAR wind retrieval accuracy. Compared with SMAP measurements, our SAR retrievals can provide fine structures of TC wind fields.

**Keywords:** synthetic aperture radar; Sentinel-1 dual-polarized imagery; wind speed retrieval; tropical cyclone; low wind





## 1. Introduction

The spaceborne synthetic aperture radar (SAR) has the capability to acquire backscatter from the ocean surface in all-weather conditions [1,2]. This imagery has been widely used to detect ocean dynamic phenomena, such as wind vectors [3,4], surface waves [5–8], surface currents [9,10], and internal waves [11,12]. In general, ocean surface roughness is positively dependent upon wind speed. SAR sensors can receive more Bragg or non-Bragg backscatter when surface wind speed increases. Based on this theory, wind speeds could be retrieved from SAR imagery [13]. With the advantages of high resolution, large coverage, and high accuracy, SAR wind retrievals have contributed to oceanic and atmospheric investigations, for example, the analysis of sea surface wind distribution [14,15], the estimation of tropical cyclone (TC) intensity, size and fullness [16,17], and the initialization of numerical models [18,19].

C-band SAR instruments in orbit, such as RADARSAT-2, Sentinel-1, and Gaofen-3, generally have a dual-polarization sensing mode. This model has the advantage of receiving signals in both co- (i.e., VV- and HH-polarization) and cross-polarization (i.e., VH- and HV-polarization), which contain abundant backscattering information to observe the ocean surface more effectively. The geophysical model function (GMF), which relates the microwave normalized radar cross section (NRCS) to the wind speed, has played a key role

in the wind retrieval from co-polarization and cross-polarization channels of SAR. To date, many empirical functions have been proposed. For co-polarized data, there is the CMOD family, including CMOD4 [20], CMOD5.N [21], CMOD7 [22], etc., originally developed for scatterometers, together with the C_SARMOD [23] and C_SARMOD2 [24] models re-tuned for SAR. For cross-polarized SAR imagery, there are C-2PO [2], C-2POD [25], C-3PO [26], and MS1A [27], and a large number of modified models, such as S1IW.NR [28], S1ER.NR [29] and MMS1A [17]. For most GMFs, the sigma nought ($\sigma_0$) is on the left side of the equation. The right side is a function of wind speed, wind direction, and the incident angle. In wind retrieval, people usually have to pre-set a group of wind speed values and then input them into a GMF to obtain a first-guess $\sigma_0$, and finally match them with SAR-measured NRCS to determine wind speeds. The difficulty of inversing GMFs makes wind retrieval less direct and leads to greater computational complexity.

According to the relationships between NRCS and wind speed, the most significant difference between co- and cross-polarization GMFs is the wind retrieval range. Since co-polarized signals saturate under high wind conditions (>25 m/s), the imagery cannot recognize the high wind area; therefore, they are often used to retrieve low-to-moderate wind speeds (<25 m/s) [29]. Meanwhile, cross-polarized acquisitions keep high sensitivity with high winds but are not good at low-wind retrieval due to the strong impact of thermal noise compared with conventional co-polarized acquisitions [27,29,30]. In addition, the wind direction parameter is another difference between co- and cross-polarization GMFs. Many studies have indicated that cross-polarized NRCS is almost insensitive to wind direction and, therefore, most cross-polarization GMFs do not require a wind direction input, making retrieval easier [31,32]. However, co-polarized NRCS is sensitive to wind direction since wind direction can change the local incident angle to influence Bragg contribution; thus, the GMFs for co-polarized images generally need a wind direction input [20,22,33].

With dual-polarized imagery, the differing sensitivity between contemporaneous co- and cross-polarized SAR signals can be advantageously combined to retrieve winds ranging from low to high as well as wind directions. Zhang et al. employed CMOD5.N and C-2POD to construct a cost function to merge information from both VV and VH channels [25]. The minimization of the cost function allows optimum estimates for wind vectors. This method performs well for the RADARSAT-2 imagery of TCs according to the comparison of retrievals with collocated QuikSCAT scatterometer winds. Mouche et al. also developed a cost function for the determination of the maximum probability to obtain a wind vector based on CMOD5.N, MS1A, and the priori information of wind vectors from ECMWF (spatial resolution is 0.125° with a time step of 3 h) [27]. The combination model performed better than the single GMF in two aspects. On the one hand, compared with the single use of the CMOD5.N model, the combination method improved the accuracy of SAR retrieval for winds larger than 25–30 m/s using the cross-polarization channel of the Sentinel-1 C-SAR sensor. On the other hand, despite noise equivalent sigma zero (NESZ) correction, the impact of NESZ remains significant at sub-swath boundaries for low wind regions. The combination model can solve this issue with the contribution of CMOD5.N and ECMWF references. Overall, excellent consistency was found between the Soil Moisture Active Passive (SMAP) winds and the results of their method, indicating that the joint use of co- and cross-polarization can indeed improve wind retrieval under TC conditions.

In this paper, a new approach was explored to retrieve ocean surface wind speed with the combination of co- and cross-polarized imageries, inspired by the well-designed empirical wave retrieval model family CWAVE [34–36], instead of a cost function. We focused on the extra wide swath (EW) mode and the interferometric wide swath (IW) mode of the Sentinel-1 A/B sensors and collected 28 TC cases with multiple wind collocations for model development and validation. Different schemes were investigated for inputs. Multiple linear regression (MLR) was utilized in modeling. A comparison was made between the proposed models and some traditional models. In addition, wind retrieval bias was assessed against the rain rate. The remaining sections of this paper are organized as

follows. Section 2 introduces the Sentinel-1 imagery, wind references, and data collocation. Section 3 presents a series of functions for the new model. In Section 4, the new models are validated with TC wind measurements and analyses. Discussion and conclusions are made in Sections 5 and 6, respectively.

## 2. Dataset Description

### 2.1. Sentinel-1 SAR Imagery

The C-SAR sensor on board the European Space Agency (ESA) Sentinel-1 A/B satellites can be operated in four different modes: the Stripmap (SM) mode, the Interferometric Wide swath (IW) mode, the Extra Wide Swath (EW) mode and the Wave (WV) mode. For TC observations, the EW mode and the IW mode are generally used to acquire extensive wind field information with the largest and second-largest swaths in four modes. Some imagery parameters are shown in Table 1. Except for these parameters, EW and IW imagery have different NESZ distributions, leading to different NRCS–wind relationships when surface wind is weak, which should be particularly considered in model development. In this study, 16 and 12 TC cases were, respectively, observed in EW and IW modes from 2016 to 2021, with dual-polarization (VV + VH). The TCs were over the Atlantic Ocean (AL), the East Pacific Ocean (EP), the Southern Hemisphere Ocean (SH), and the West Pacific Ocean (WP). Sentinel Application Platform (SNAP) 7.0 software was utilized for GRD border noise removal, thermal noise removal, and imagery calibration.

**Table 1.** Some imagery parameters of the Sentinel-1 EW and IW modes.

| Sensor Mode | Ground Swath (km) | Spatial Resolution [1] (m) | Incident Angle (°) | Sub-Swaths Number |
|---|---|---|---|---|
| EW | 410 | $30 \times 40$ | 20–47 | 5 |
| IW | 250 | $5 \times 20$ | 31–46 | 3 |

[1] The spatial resolution is range $\times$ azimuth.

### 2.2. SFMR Wind and Rainfall Measurements

The National Oceanic and Atmospheric Administration (NOAA) Stepped-Frequency Microwave Radiometer (SFMR) on board reconnaissance aircraft can measure the brightness temperature from the sea surface at six C-band frequencies along the flight track over a storm [37]. Ocean surface wind speeds are retrieved according to a function between the wind speed and surface emissivity. Rain rate retrieval is based on the relationship between microwave absorption and rain rate. The SFMR wind speed measurements have an RMSE of ~3.9 m/s compared with the dropsonde measurements [38]. SFMR wind speeds and rain rate with a spatial resolution of 0.01° have been used as collocations in the establishment and evaluation of the SAR wind and rainfall retrieval models [39,40]. Our study used data for model development.

### 2.3. SMAP Wind Measurements

The L-band radiometer on board the SMAP satellite can measure the bright temperature to retrieve the ocean surface wind speed without being affected by rainfall [41,42]. Its 1000 km swath provides it with a chance to make a complete observation of a storm. Remote Sensing Systems (www.remss.com/missions/smap/ (accessed on 21 March 2022)) provided the SMAP Final Wind Speed products and considered the equivalent neutral winds at a 10 m height above the ocean surface. Their grid spacing was 0.25° × 0.25°. For winds greater than 25 m/s, a consistency was found between SMAP and SFMR measurements, with a bias of 0.5 m/s and a standard deviation of ~3 m/s [38]. In addition, the evaluation of size values proved that the SMAP winds could be used for TC wind field monitoring and modeling [43]. In this paper, we used the SMAP Final Wind Speed products as an independent data source for validation.

### 2.4. Dropsonde Wind Vector Measurements

To validate the proposed wind retrieval models, we collected NOAA Hurricane Research Division (HRD) dropsonde data. Dropsondes were released from the aircraft along the flight track and drifted down on a parachute, measuring vertical profiles of pressure, temperature, humidity, and wind as they fell. At a 10 m height, the accuracy of the wind speed measurement was about 0.5–2.0 m/s [44]. In this work, only sea surface wind vector measurements were used whose pressure level of observation was 1070.0 hPa. They acquired over 8 TCs. There were 11 observations collocated with EW mode SAR images and 12 observations collocated with IW mode SAR images. The time difference between dropsonde observations and SAR images was less than 0.5 h.

### 2.5. ECMWF ERA5 Wind Vectors

ERA5 is the fifth-generation ECMWF atmospheric reanalysis of the global climate, covering the period from 1950 to the present. Based on data assimilation, ERA5 combines model data with observations from across the world into a global dataset with an hourly temporal resolution and a spatial resolution of $0.25° \times 0.25°$ [45]. ECMWF wind vectors have been used as references in investigations of SAR wind retrieval with both signal and dual polarizations. In this study, all wind direction values were collected from ERA5, except a case study on Tropical Storm Karl (2016). In addition, most wind speed values lower than 15 m/s were from ERA5 in model development as a supplement to SFMR measurements.

### 2.6. H*Wind Data

H*Wind is an integrated system that provides a view of the strength and extent of hurricane wind fields according to the input of all available surface weather observations with respect to the storm and is used operationally to improve the damage assessment of hurricane intensity [46]. In general, the total uncertainty of the H*Wind products in hurricanes is approximately 6% near the storm center, increasing to nearly 13% near the tropical storm force wind radius [47]. In this paper, a case study of Tropical Storm Karl was carried out with the wind vectors provided by Risk Management Solutions H*Wind. The spatial resolution was $0.01° \times 0.01°$.

### 2.7. Collocation

To construct the dataset, the Sentinel-1 imagery of TCs was collocated with the wind data introduced above. There were 13 SFMR and 16 SMAP collocations. All SAR imageries were matched with ERA5. The time differences between SAR and SFMR were controlled within 0.5 h. The time difference between SAR and SMAP was just a few minutes. The time difference between SAR and ERA5 was within 1 h. According to the best tracks provided by the National Hurricane Center (NHC) and the Joint Typhoon Warning Center (JTWC), the TC intensities of different matches are shown with the Saffir–Simpson hurricane wind scale in Figure 1. For Tropical Storm Karl, the time difference between SAR and H*Wind was about 1 h and 37 min. According to the TC motion vector estimated from the best track and the time difference, the locations of H*Wind were shifted to match SAR imagery using the method presented in [28].

To note, a SAR image of Hurricane Michael observed at 23:44 UTC on 9 October 2018 could be matched both with SFMR and SMAP (Figure 2), where the storm eye was obvious. Figure 2a,b are VV- and VH-polarized images from the EW mode, respectively. The white and black lines crossing the storm eye are the tracks of flights belonging to the Air Force Reconnaissance (AFRC) and NOAA, respectively. Figure 2c is the collocated SMAP final wind with a maximum value of 53.69 m/s. Figure 3 shows an example of IW mode dual-polarized imagery acquired over Hurricane Irma at 9:07 UTC on 29 October 2017. The white line is the AFRC flight track.

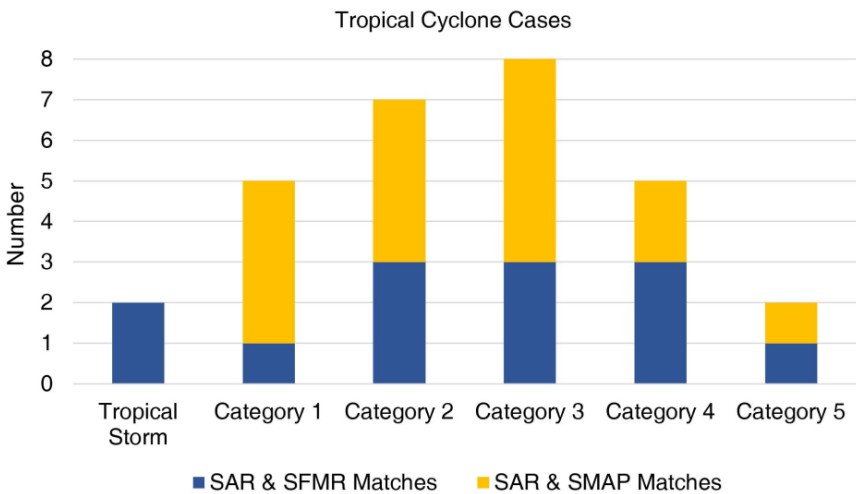

**Figure 1.** Intensities and numbers of the TC cases observed by Sentinel-1 with two main wind matches. The intensities were represented by the Saffir–Simpson hurricane wind scale and estimated from the best tracks provided by NHC and JTWC.

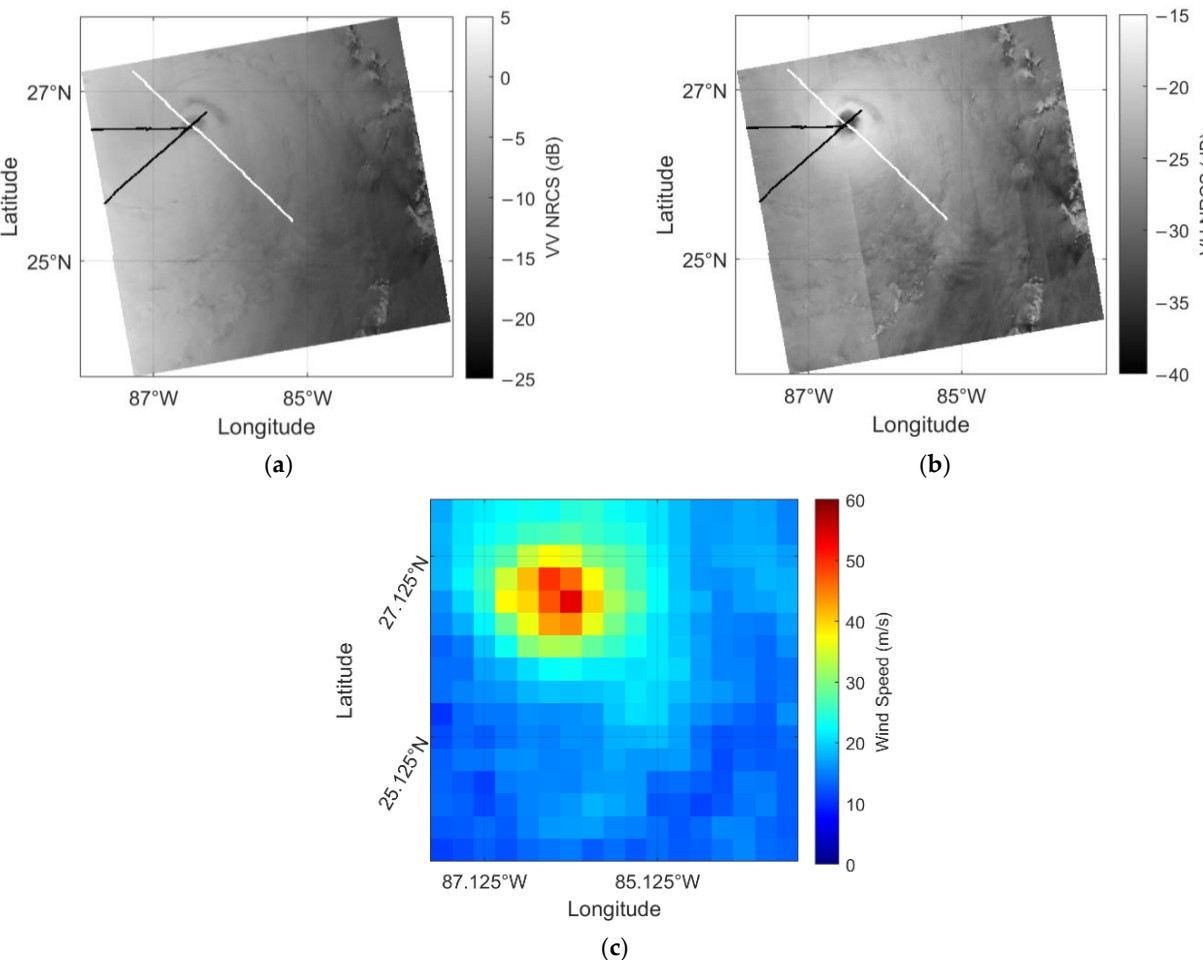

**Figure 2.** A collocation example of Hurricane Michael. (**a**) Sentinel-1 EW mode VV-polarized image acquired on 23:44 UTC 9 October 2018; (**b**) Contemporaneous VH-polarized image; and (**c**) Matched SMAP final winds measured on 23:36 UTC 9 October 2018. In (**a**,**b**), white and black lines crossing the storm eye are the tracks of flights from AFRC and NOAA, respectively. The SFMR observations along the tracks are within a 0.5 h time difference from Sentinel-1.

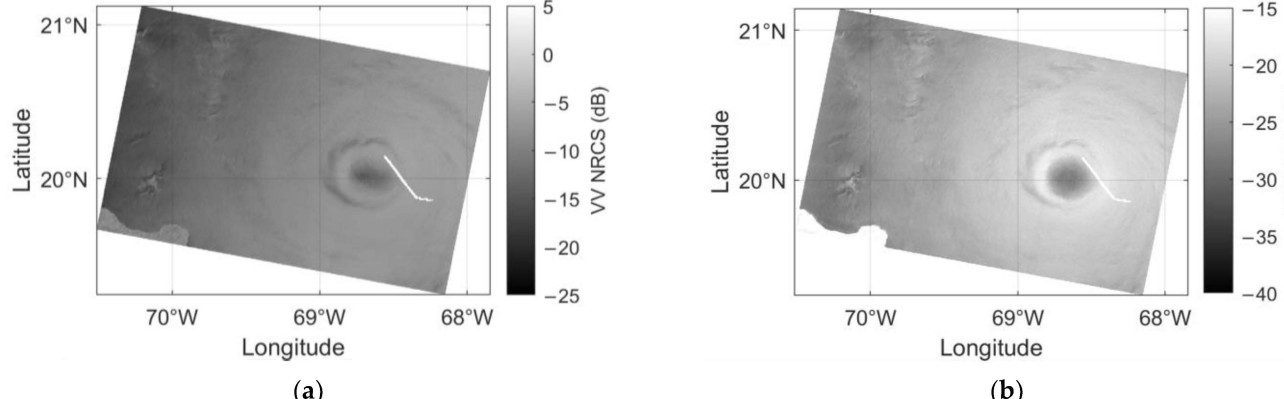

**Figure 3.** A collocation example of Hurricane Irma. (**a**) Sentinel-1 IW mode VV-polarized image acquired on 9:07 UTC 29 October 2017; (**b**) Contemporaneous VH-polarized image. White lines show the AFRC flight track. The SFMR observations along the track are within a 0.5 h time difference from Sentinel-1.

## 3. Model Development

After data collocation, the distribution of NRCS was compared with the incident angle and wind speed (Figure 4). The winds of the unfilled points were measured using SFMR. Since under TC conditions, most of these winds were larger than 15 m/s, the low winds of the filled points were collected from ERA5 reanalysis as a supplement for data fitting in an entire wind speed range. Figure 4a–d shows the VV- and VH-polarization for the EW and IW modes, respectively. Low winds have a strong linear correlation with VV NRCS. The range of VV NRCS is wide enough to distinguish specific wind speeds. However, for the VH channel, the signals of low winds are seriously affected by NESZ (shown with black curves in Figure 4b,d), making the retrieval difficult. These are the main reasons why VV-polarized imagery performed well in the retrieval of low wind speed, but VH-polarized imagery did not. Interestingly, the opposite happened in moderate and high winds. The linear correlation between NRCS and wind speed was stronger in VH imagery than in VV imagery and weakly impacted by the incident angle and NESZ. So, if signals from VV- and VH-polarization channels are combined for model development, the new model could take advantage of both channels.

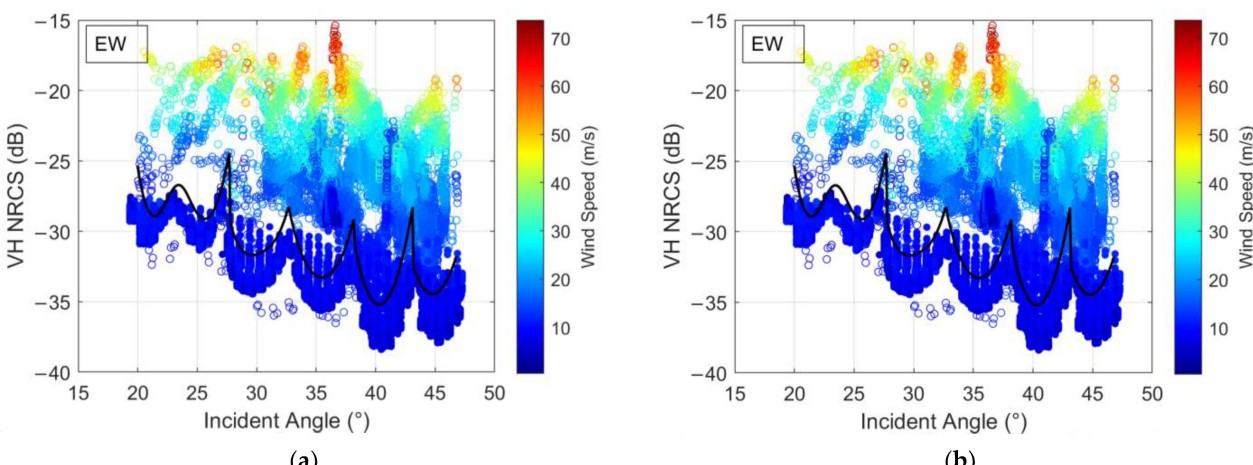

**Figure 4.** *Cont.*

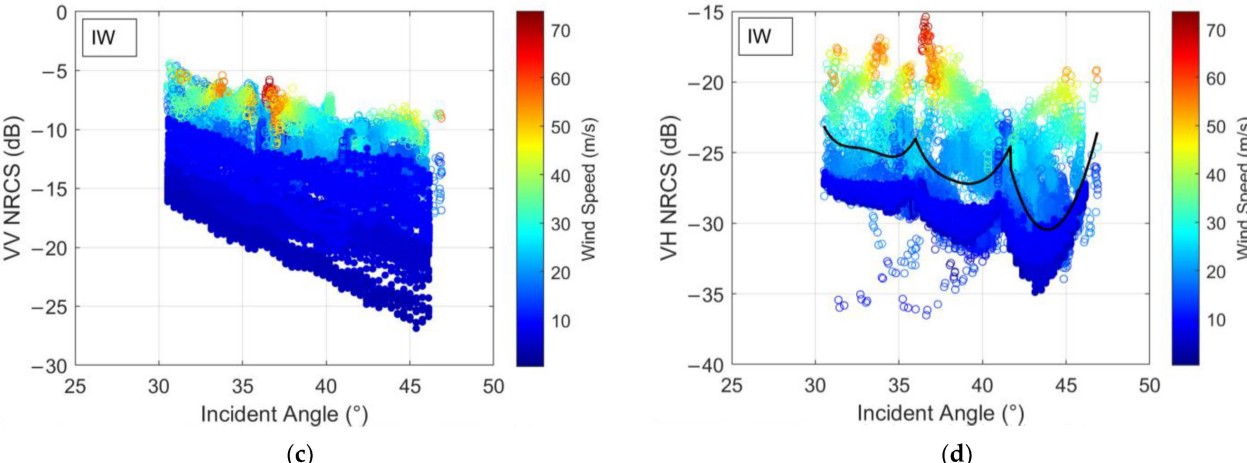

**Figure 4.** Comparison between NRCS, incident angle and wind speed for (**a**) EW mode VV-polarization; (**b**) EW mode VH-polarization; (**c**) IW mode VV-polarization; (**d**) IW mode VH-polarization. The winds of the unfilled points were measured using SFMR. The winds of the filled points are from the ERA5 reanalysis. Black curves stand for NESZ.

Inspired by the CWAVE family of the SAR empirical wave retrieval models [34–36], we present a new approach, i.e., Equation (1), to establish the model. Sentinel-1 dual-polarized signals were combined as fitted data. The output is the wind speed at 10 m height ($U_{10}$). There are four inputs: VH NRCS ($\sigma_0^{VH}$), incident angle ($\theta$), VV NRCS ($\sigma_0^{VV}$), and wind direction ($\varphi$). To note, the wind direction is the angle between the real and azimuth directions. We used MLR to automatically decide which parameter was important for wind retrieval and expressed the relationship by determining coefficients. Three different schemes were tested for model inputs. We named them Model 1 (inputs: $\sigma_0^{VH}$, $\theta$), Model 2 (inputs: $\sigma_0^{VH}$, $\theta$, $\sigma_0^{VV}$) and Model 3 (inputs: $\sigma_0^{VH}$, $\theta$, $\sigma_0^{VV}$, $\varphi$).

$$U_{10} = \sum_{i,j=1}^{n} A_{i,j} X_i X_j + \sum_{i=1}^{n} A_i X_i + A_0, \tag{1}$$

where $A_0$, $A_i$ and $A_{i,j}$ are coefficients. For their values, please refer to Table A1. $X_i$ and $X_j$ are inputs. $j \geq i$. $X_1 = \sigma_0^{VH}$. $X_2 = \theta$. $X_3 = \sigma_0^{VV}$. $X_4 = \varphi$. $n$ = 2, 3, 4 for Model 1, 2, and 3, respectively. Wind retrieval models were separately developed for the EW mode and the IW mode. Then, for evaluation, the new models were used to simulate wind speeds from the fitted data. Figures 5 and 6 show the comparison of the simulations and wind references for two modes.

After MLR fitting, an additional linear regression was utilized to correct the bias, especially for the retrieval of high winds. The final retrieval was $U_{10}^{final} = a U_{10}^{b}$. $a$ and $b$ are coefficients. The values of $a$ and $b$ are listed in Table A2. Finally, the new models were evaluated again through an error calculation based on the fitted data, including the bias (retrievals minus fitted data), Pearson correlation coefficient (Cor), and root mean squared error (RMSE). The calculation methods of Cor and RMSE are given in Equations (2) and (3). The results are illustrated in Table 2. In general, the error reduced if a proposed model had more inputs, indicating that the combination of VV- and VH-polarized signals, together with wind direction, have the potential to improve retrieval accuracy.

$$\text{Cor} = \frac{\text{cov}\left(U_{10}^{final}, \ U_{10}^{reference}\right)}{\text{std}\left(U_{10}^{final}\right)\text{std}\left(U_{10}^{reference}\right)}, \tag{2}$$

$$\text{RMSE} = \sqrt{\frac{\sum_{i=1}^{N}\left(U_{10,i}^{final} - U_{10,i}^{reference}\right)^2}{N}} \tag{3}$$

where $U_{10}^{\text{final}}$ is the SAR-simulated wind speed and $U_{10}^{\text{reference}}$ is the reference wind speed. $\text{cov}(U_{10}^{\text{final}}, U_{10}^{\text{reference}})$ stands for the covariance of them. $\text{std}(U_{10}^{\text{final}})$ and $\text{std}(U_{10}^{\text{reference}})$ are the standard deviations of $U_{10}^{\text{final}}$ and $U_{10}^{\text{reference}}$.

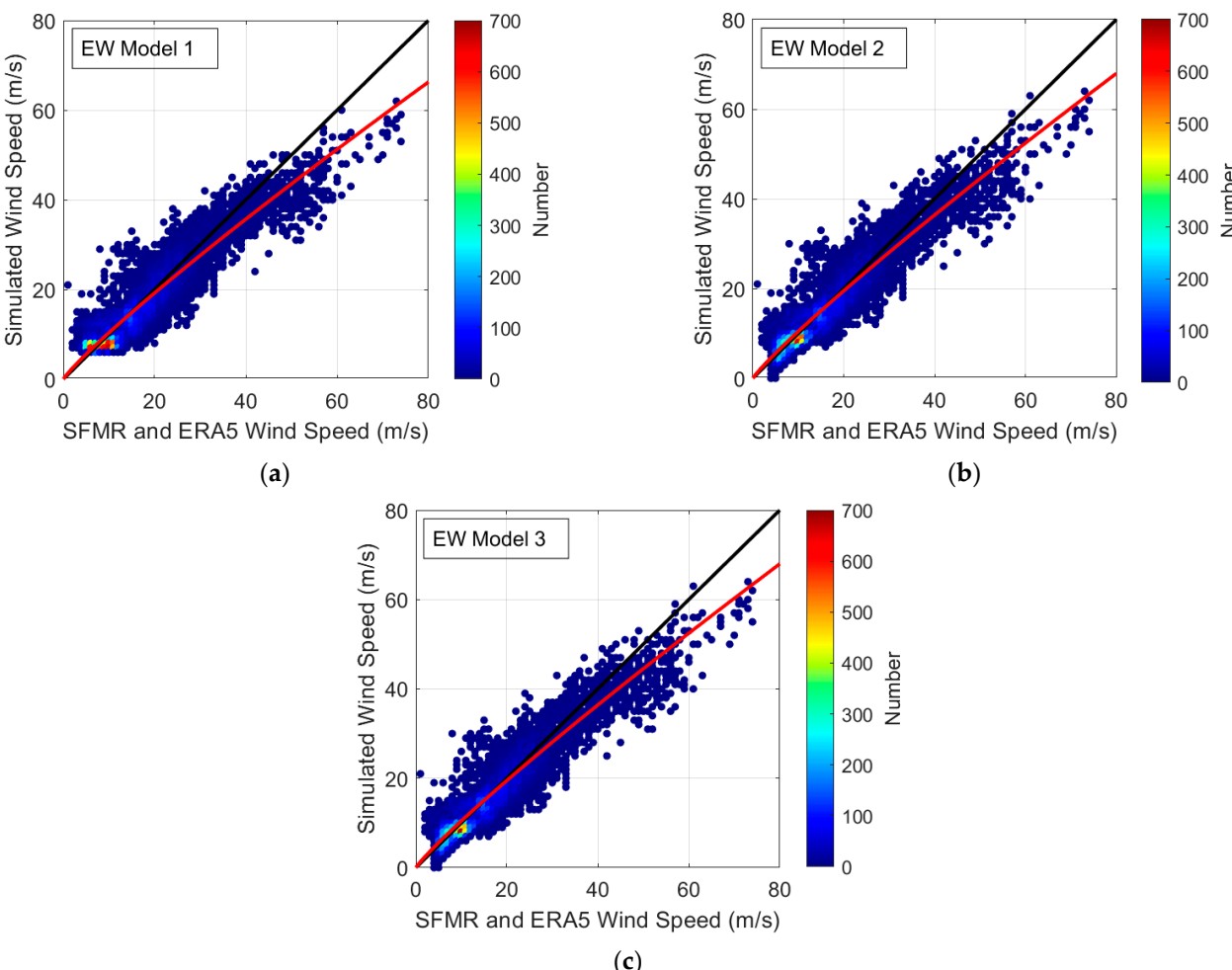

**Figure 5.** Comparison of the model-simulated and fitted wind speeds for different models: (**a**) EW Model 1; (**b**) EW Model 2; (**c**) EW Model 3. Red curve stands for the additional linear regression.

**Table 2.** Evaluation of the new models based on fitted data.

| Model | Bias (m/s) | | | Cor | | | RMSE (m/s) | | |
|---|---|---|---|---|---|---|---|---|---|
| | <10 m/s | ≥10 m/s | All | <10 m/s | ≥10 m/s | All | <10 m/s | ≥10 m/s | All |
| EW Model 1 | 0.75 | −0.32 | 0.13 | 0.04 | 0.92 | 0.93 | 2.66 | 4.09 | 3.57 |
| EW Model 2 | 0.42 | −0.17 | 0.07 | 0.40 | 0.92 | 0.94 | 2.41 | 3.79 | 3.30 |
| EW Model 3 | 0.38 | −0.13 | 0.08 | 0.40 | 0.92 | 0.94 | 2.42 | 3.77 | 3.28 |
| IW Model 1 | 1.24 | −0.86 | 0.0045 | 0.27 | 0.90 | 0.91 | 2.77 | 5.18 | 4.36 |
| IW Model 2 | 0.51 | −0.34 | 0.0097 | 0.67 | 0.90 | 0.93 | 2.59 | 4.30 | 3.69 |
| IW Model 3 | 0.24 | −0.16 | 0.0049 | 0.64 | 0.91 | 0.94 | 2.34 | 4.05 | 3.45 |

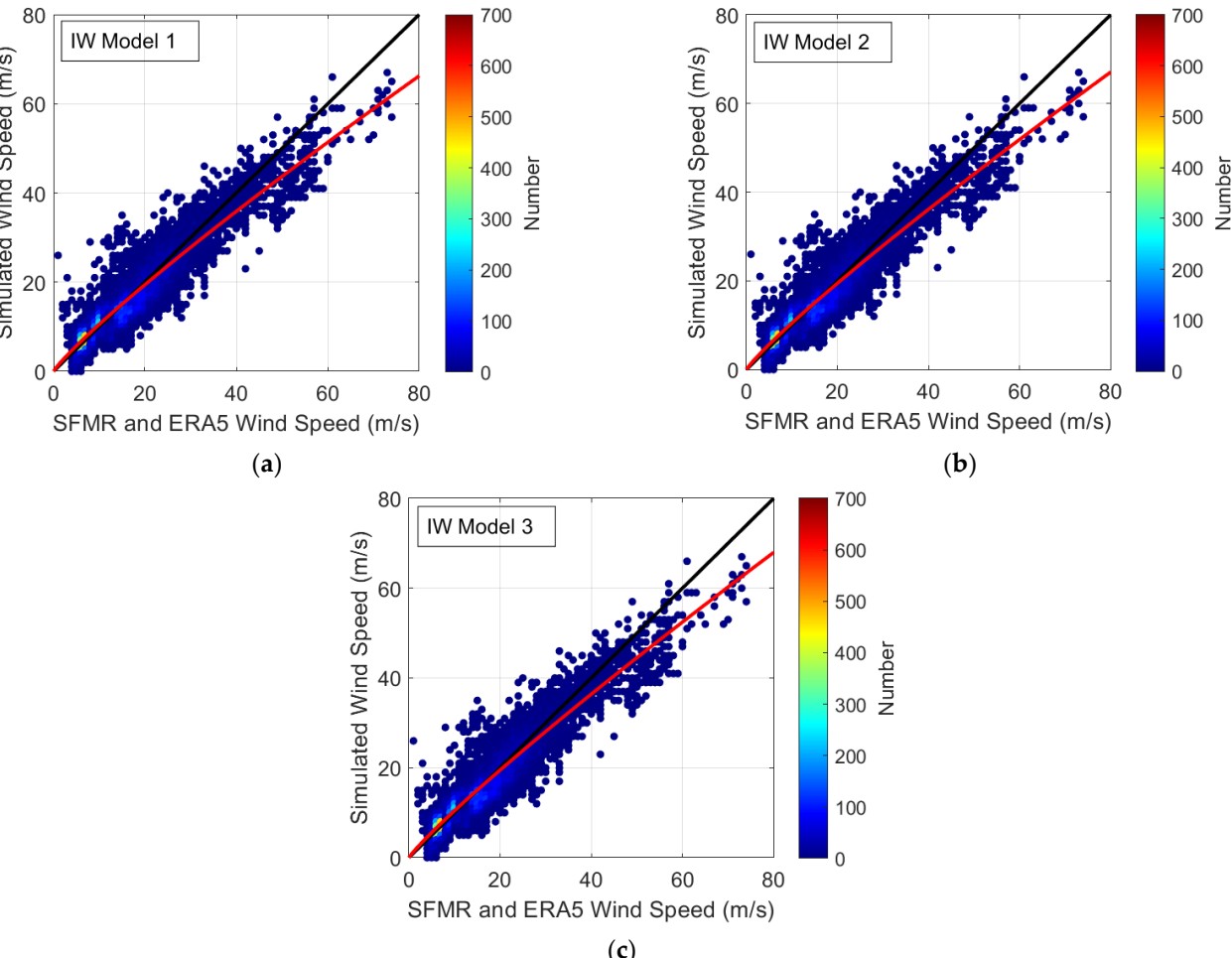

**Figure 6.** Comparison of the model-simulated and fitted wind speeds for different models: (**a**) IW Model 1; (**b**) IW Model 2; (**c**) IW Model 3. Red curve stands for the additional linear regression.

## 4. Validation and Comparison

### 4.1. Validation with SMAP Winds

To evaluate the accuracy of the proposed models, validation was carried out in this section. Wind speeds were retrieved via different models from 16 TC imageries observed using the Sentinel-1 EW and IW modes. Then, retrievals were compared with the collocated SMAP wind speeds, ensuring the objectivity of validation. ERA5 wind directions were used as inputs in EW Model 3 and IW Model 3. The sample number distributions of SMAP wind speeds and ERA5 wind directions are shown in Figure 7. The maximum wind was up to about 70 m/s. Wind directions were distributed from 0 to 360 degrees.

Models were validated separately for the Sentienl-1 EW and IW modes. For the EW mode, wind speeds were retrieved by EW Model 1, 2, 3, MMS1A, and S1EW.NR. MMS1A and S1EW.NR are piecewise functions. The MMS1A model was modified from MS1A through a correction of SFMR data [17]. The S1EW.NR model was fitted from ERA5 wind speeds [29]. For wind retrieval, the two models just required VH NRCS and incident angle inputs.

Comparisons are shown in Figure 8. According to the scatters, error statistics were carried out for winds lower than 10 m/s, higher than 10 m/s, and for all winds. The results are illustrated in Table 3. Overall, EW Model 2 performed best. Better consistencies could be found between the results of the proposed models and SMAP winds. The main reason was the modeling method: MLR was able to develop a single function that could make retrievals change more smoothly in adjacent sub-swaths of SAR imagery. However, this is difficult to realize with a piecewise function. In addition, S1EW.NR has a more serious

underestimation. Compared with the S1EW.NR model, the MS1A model, was closer to the new models. It showed another factor that affects model performance: the fitted data. To note, all the tested models underestimated winds higher than 30 m/s. This issue was induced by the high wind difference between SFMR, ERA5, and SMAP.

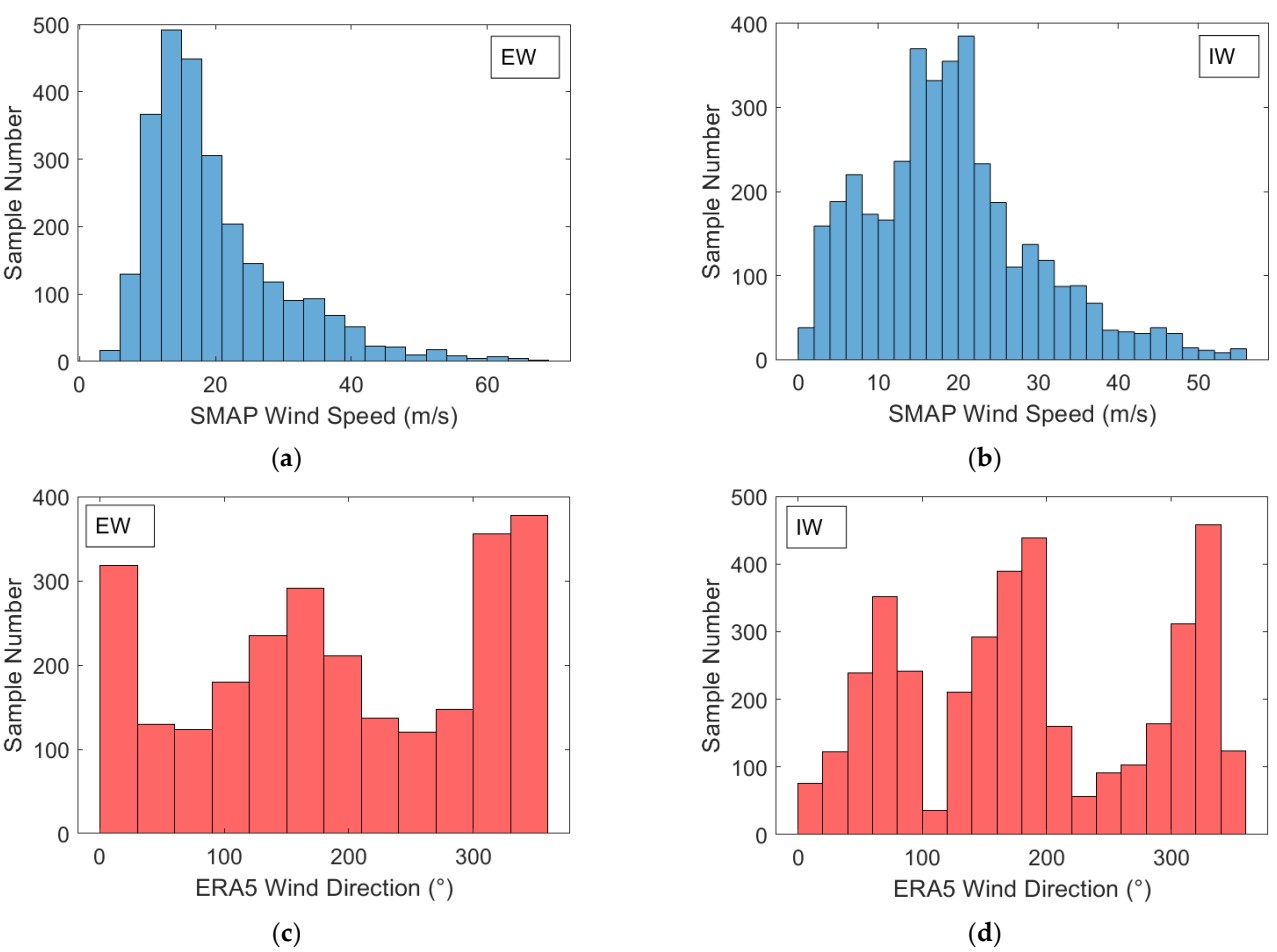

**Figure 7.** Sample number distributions of the SMAP wind speeds used for (**a**) EW mode; (**b**) IW mode; the ERA5 wind directions used for (**c**) EW mode; (**d**) IW mode.

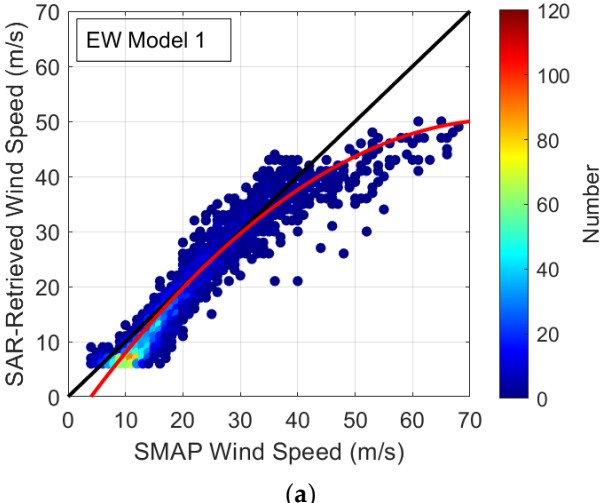
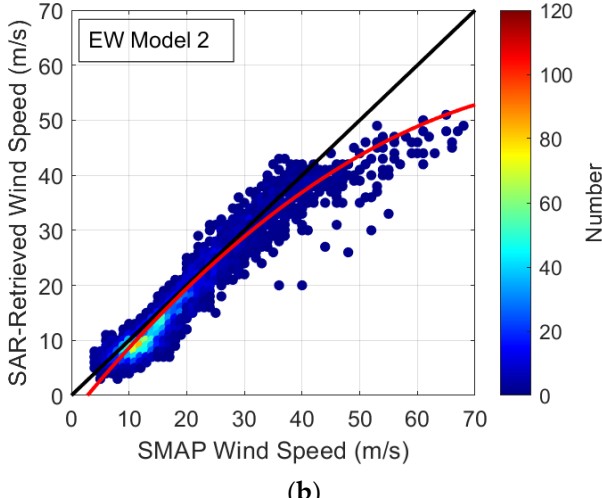

**Figure 8.** *Cont.*

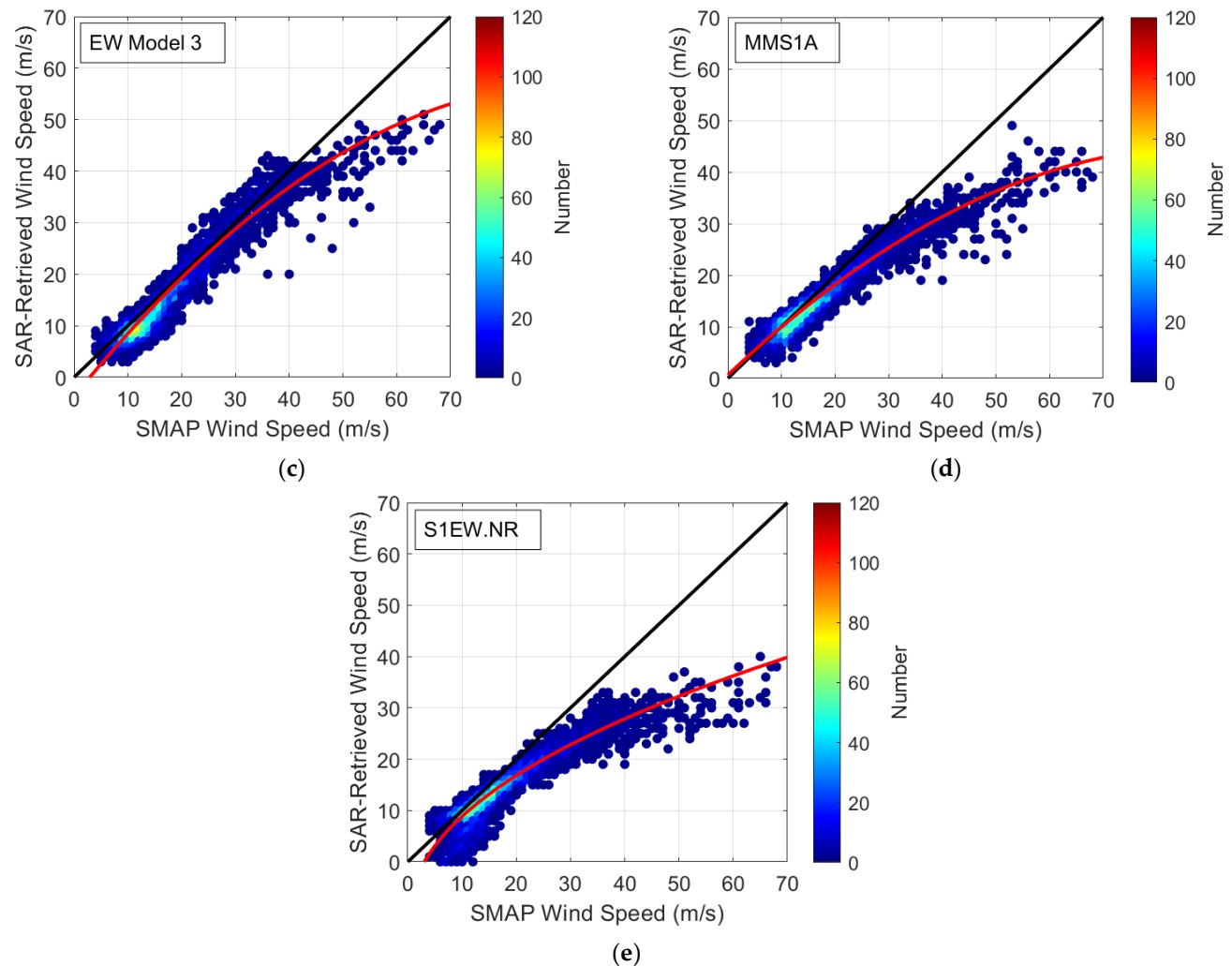

**Figure 8.** Comparison between SMAP wind speed measurements and SAR-retrieved wind speeds based on (**a**) EW Model 1; (**b**) EW Model 2; (**c**) EW Model 3; (**d**) MMS1A; (**e**) S1EW.NR. Red curve stands for the variation trend.

**Table 3.** Validation results of the proposed EW Model 1, 2, 3, and traditional MMS1A and S1EW.NR models. Retrievals were compared with SMAP wind speeds.

| Model | Bias (m/s) | | | Cor | | | RMSE (m/s) | | |
|---|---|---|---|---|---|---|---|---|---|
| | <10 m/s | ≥10 m/s | All | <10 m/s | ≥10 m/s | All | <10 m/s | ≥10 m/s | All |
| EW Model1 | −0.70 | −1.47 | −1.36 | 0.26 | 0.94 | 0.95 | 1.76 | 3.60 | 3.37 |
| EW Model2 | 0.10 | −1.53 | −1.27 | 0.39 | 0.95 | 0.96 | 1.74 | 3.30 | 3.10 |
| EW Model3 | −0.16 | −1.64 | −1.41 | 0.41 | 0.95 | 0.96 | 1.79 | 3.34 | 3.15 |
| MMS1A | 0.71 | −2.37 | −1.88 | 0.44 | 0.95 | 0.96 | 2.01 | 4.30 | 4.02 |
| S1EW.NR | −0.77 | −3.89 | −3.40 | 0.21 | 0.92 | 0.92 | 3.21 | 5.92 | 5.58 |

For the Sentienl-1 IW mode, wind speeds were retrieved via IW Model 1, 2, 3, MMS1A, and S1IW.NR. The S1IW.NR model is linear and has a power piecewise function, which is established with SFMR and SMAP data [28]. Figure 9 shows the comparisons of the retrievals and SMAP winds. According to the scatters, the performance metrics are illustrated in Table 4. Just like the EW situation, when compared with the two traditional models, the proposed models had smaller RMSE values. The retrievals of high winds were smaller than SMAP data. Their reasons are the same as the EW situation. In addition, since MMS1A was mainly designed for the EW mode, it had the largest error for winds

lower than 10 m/s, indicating that one GMF is unable to fit low winds for both EW and IW modes. This is because the low VH NRCS is seriously affected by different thermal noises for different modes.

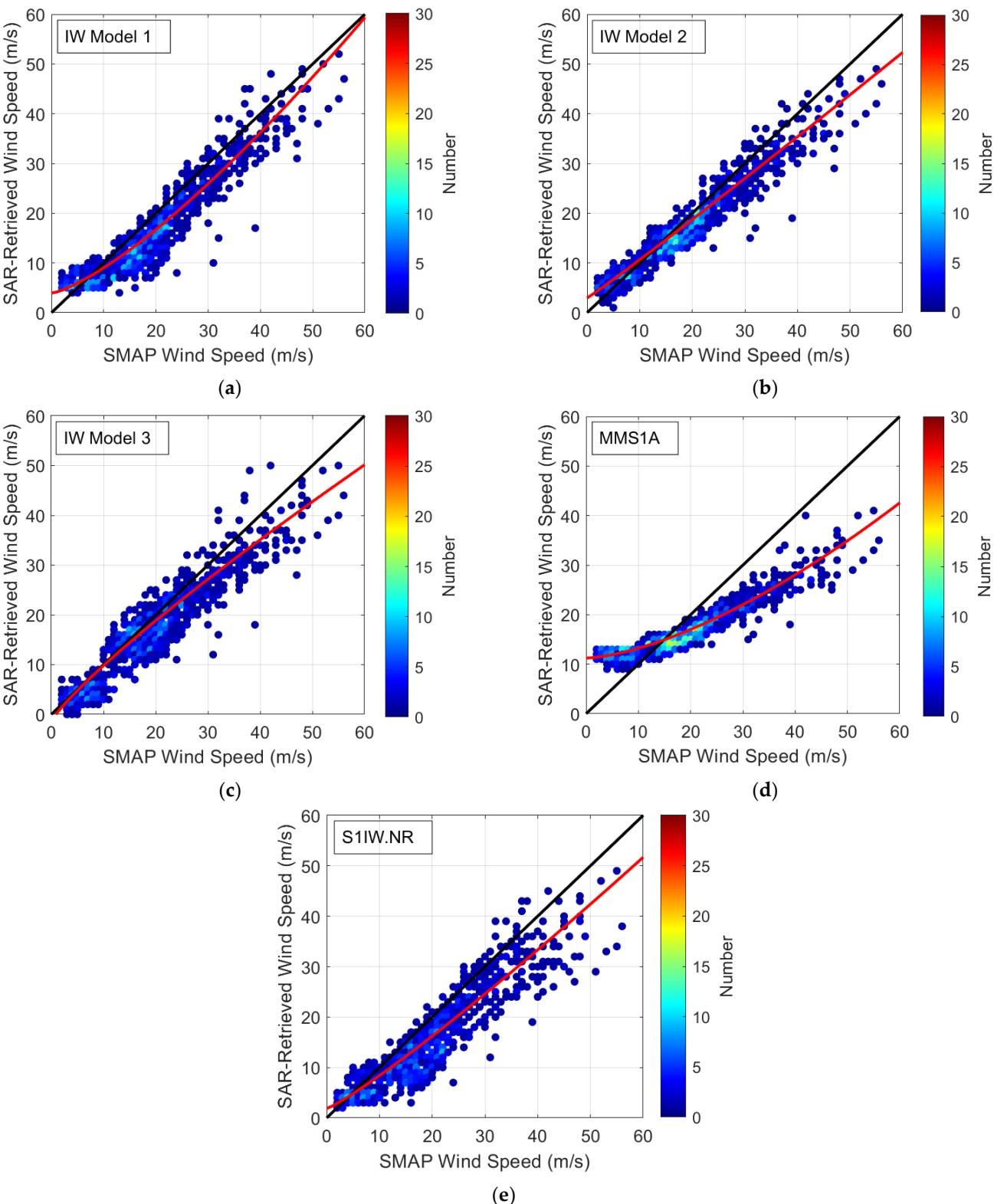

**Figure 9.** Comparison between SMAP wind speed measurements and the SAR-retrieved wind speeds based on (**a**) IW Model 1; (**b**) IW Model 2; (**c**) IW Model 3; (**d**) MMS1A; (**e**) S1IW.NR. Red curve stands for the variation trend.

**Table 4.** Validation results of the proposed IW Model 1, 2, 3, and traditional MMS1A and S1IW.NR models. Retrievals were compared with SMAP wind speeds.

| Model | Bias (m/s) | | | Cor | | | RMSE (m/s) | | |
|---|---|---|---|---|---|---|---|---|---|
| | <10 m/s | ≥10 m/s | All | <10 m/s | ≥10 m/s | All | <10 m/s | ≥10 m/s | All |
| IW Model1 | 1.18 | −3.14 | −2.26 | 0.27 | 0.93 | 0.94 | 2.71 | 4.55 | 4.24 |
| IW Model2 | 1.36 | −1.79 | −1.14 | 0.58 | 0.94 | 0.96 | 2.43 | 3.55 | 3.35 |
| IW Model3 | −1.07 | −1.73 | −1.60 | 0.47 | 0.89 | 0.93 | 2.56 | 4.39 | 4.08 |
| MMS1A | 6.45 | −4.04 | −1.88 | −0.01 | 0.93 | 0.93 | 6.96 | 6.03 | 6.22 |
| S1IW.NR | 0.12 | −4.23 | −3.33 | 0.44 | 0.89 | 0.92 | 2.44 | 5.90 | 5.38 |

### 4.2. Validation with Dropsonde Winds

An experiment was carried out to validate the proposed models with the winds measured using Dropsonde. The wind directions used in Model 3 were acquired from Dropsonde wind vectors. Comparisons are shown in Figure 10. The EW and IW Models 1, 2, and 3 are the proposed models. The MMS1A, S1EW.NR and S1IW.NR are traditional models. Tables 5 and 6 show the validation results for the EW and IW modes, respectively. On average, the new models performed better than the traditional models. However, due to a small number of samples, this improvement was not obvious from Models 1 to 3.

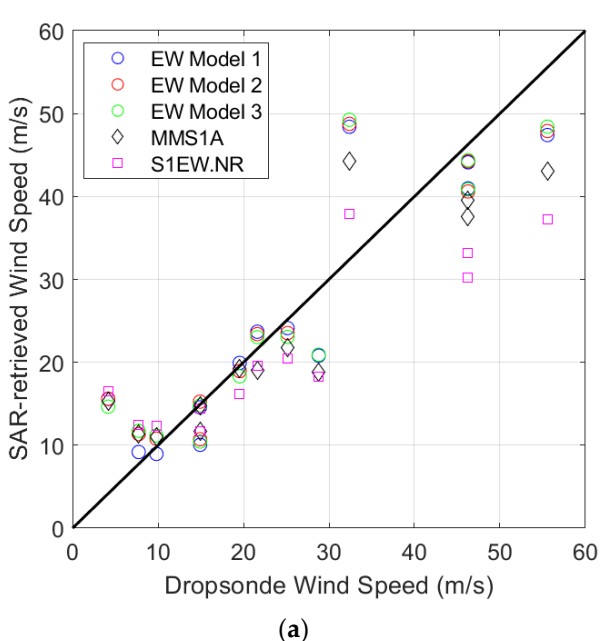
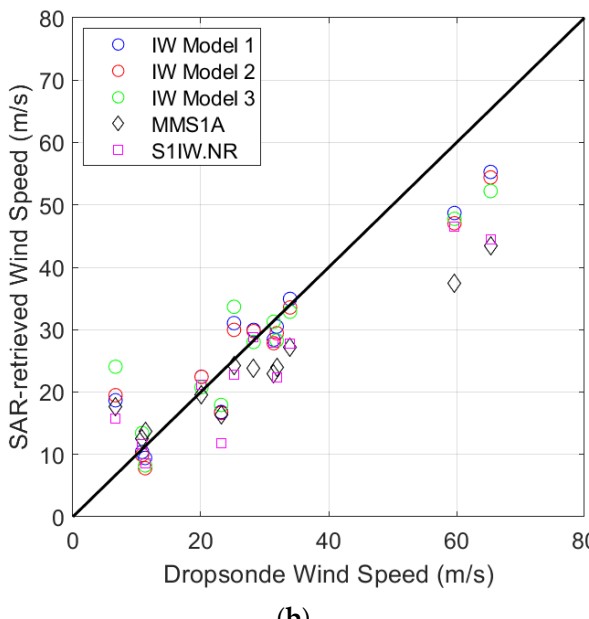

(**a**) (**b**)

**Figure 10.** Comparison of Dropsonde measurements with wind speeds retrieved from Sentinel-1 (**a**) EW mode images; (**b**) IW mode images, respectively.

**Table 5.** Validation results of the proposed EW Model 1, 2, 3, and traditional MMS1A and S1EW.NR models. Retrievals were compared with Dropsonde wind speeds.

| Model | Bias (m/s) | Cor | RMSE (m/s) |
|---|---|---|---|
| EW Model 1 | 0.05 | 0.90 | 6.69 |
| EW Model 2 | 0.37 | 0.90 | 6.76 |
| EW Model 3 | 0.30 | 0.90 | 6.71 |
| MMS1A | −1.52 | 0.90 | 7.26 |
| S1EW.NR | −3.60 | 0.88 | 9.37 |

**Table 6.** Validation results of the proposed IW Model 1, 2, 3, and traditional MMS1A and S1IW.NR models. Retrievals were compared with Dropsonde wind speeds.

| Model | Bias (m/s) | Cor | RMSE (m/s) |
|---|---|---|---|
| IW Model 1 | −0.93 | 0.95 | 6.22 |
| IW Model 2 | −1.62 | 0.94 | 6.75 |
| IW Model 3 | −0.77 | 0.91 | 7.88 |
| MMS1A | −5.42 | 0.96 | 10.60 |
| S1IW.NR | −4.85 | 0.94 | 9.03 |

### 4.3. A Case Study of Hurricane Michael

In this study, Hurricane Michael (2018) is very special. This is because it is the only sample whose SAR image was collocated with both SFMR and SMAP measurements, making it possible to compare our retrievals with winds and different grid spacings. Based on EW Model 3, the surface winds of Hurricane Michael were retrieved, as shown in Figure 11a. It had a grid spacing of 1 km. The storm eye, eyewall, and rain bands could be identified easily. This is the advantage of our model. It can provide wind fields with small structures. Figure 11b is the collocated SMAP wind map. Due to a 0.25°-grid-spacing, it looks much coarser. There are no storm eyes and other fine structures.

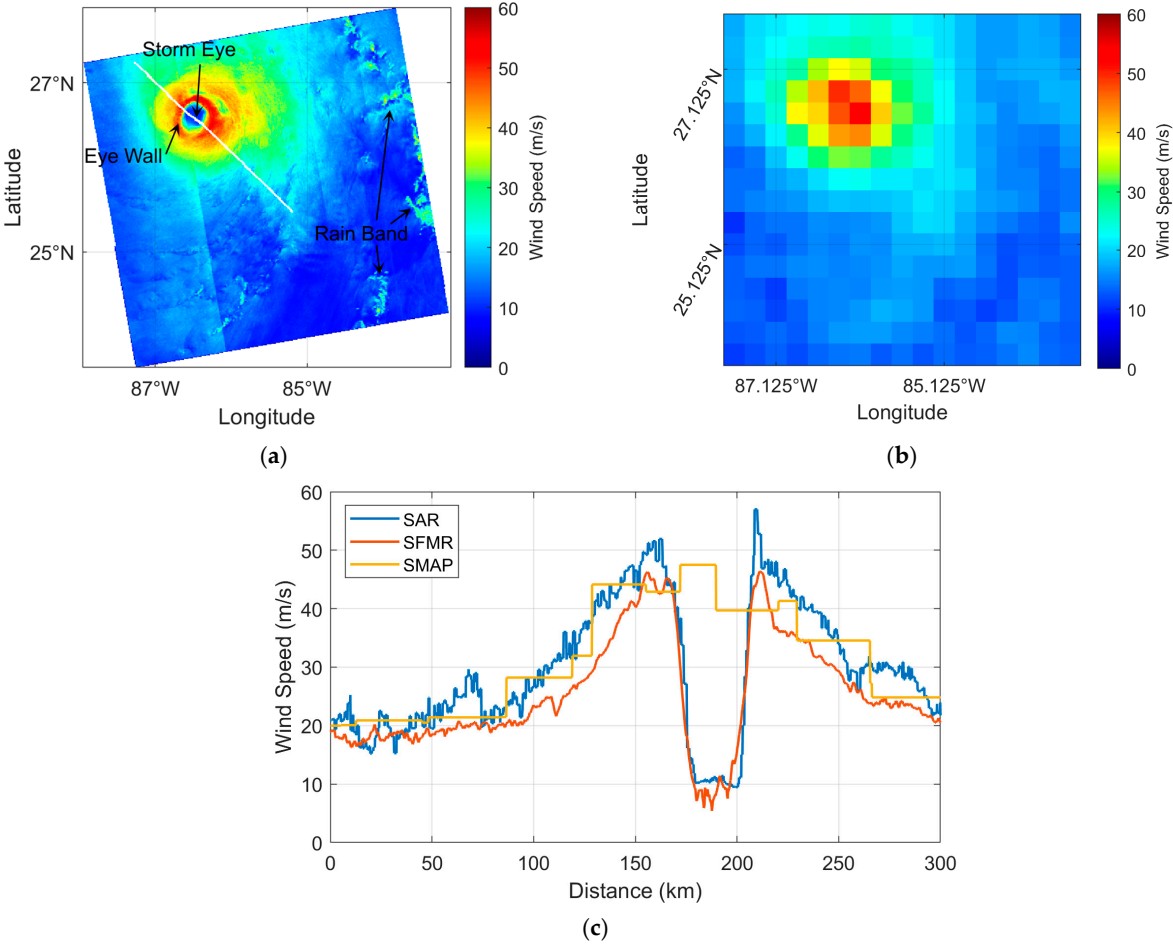

**Figure 11.** Comparison of the SAR, SFMR and SMAP wind measurements in a case study on Hurricane Michael. (**a**) Surface wind speeds retrieved via EW Model 3 from Sentinel-1 image. (**b**) Collocated SMAP wind map. (**c**) Wind comparison along the SFMR observation track (i.e., the white line in (**a**)). The grid spacing of SAR and SFMR winds was about 1 km. The grid spacing of SMAP winds was 0.25°.

In Figure 11a, the white line crossing the storm eye represents the AFRC flight track, on which the winds were measured by SFMR. The time difference between the SFMR measurements and the SAR image was less than 0.5 h. To further show the difference between the three wind products, winds along the track were compared in Figure 11b. SFMR and SAR retrievals had the same grid spacing; however, SFMR wind values were slightly lower than SAR. They could observe very low wind speeds in the storm center and radial wind profile. In the region from the eye wall to the outer region, although the distribution of SMAP data was similar to SAR and SFMR, it could not show low winds in the center and the location of maximum wind speed when facing a 50 km-diameter storm eye.

### 4.4. A Case Study of Tropical Storm Karl

A case study of Tropical Storm Karl was carried out to statistically and visually compare the retrieval differences between the proposed EW Model 1, 2, 3, MMS1A, and S1EW.NR and especially investigate the extent of thermal noise's impact on retrieval. The Sentinel-1 VV- and VH-polarized imageries, as shown in Figure 12a,b, were acquired at 22:22 UTC 29 on September 2016. Thermal noise had no impact on the VV-polarized image. However, for the VH-polarized image, the impact was obvious in low wind regions and the leftmost sub-swath 1. The collocated H*Wind data are shown in Figure 12c. Its time was 24:00 UTC on 29 September 2016. The retrievals of EW Model 1, 2, 3, MMS1A, and S1EW.NR models are shown in Figure 12d–i.

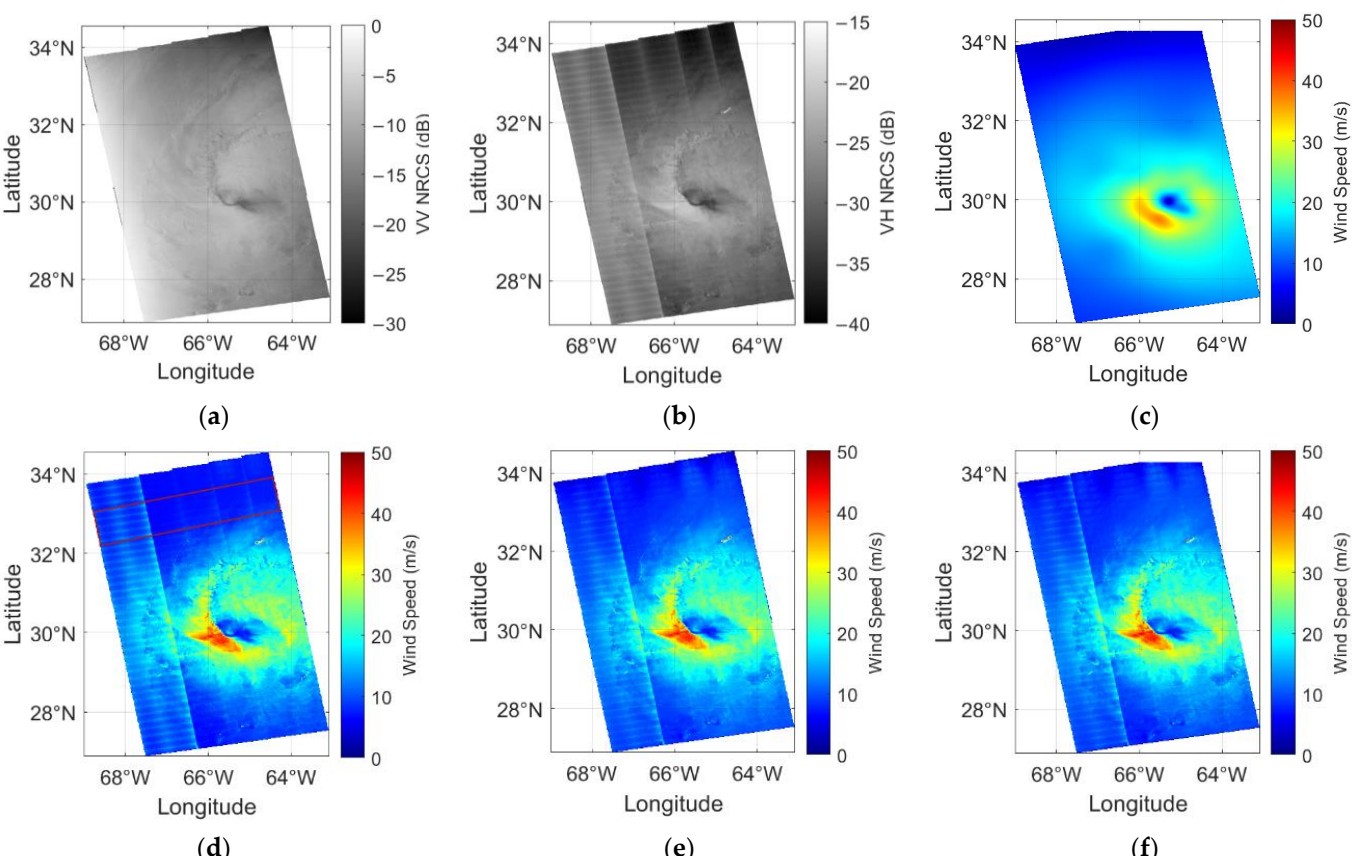

**Figure 12.** *Cont.*

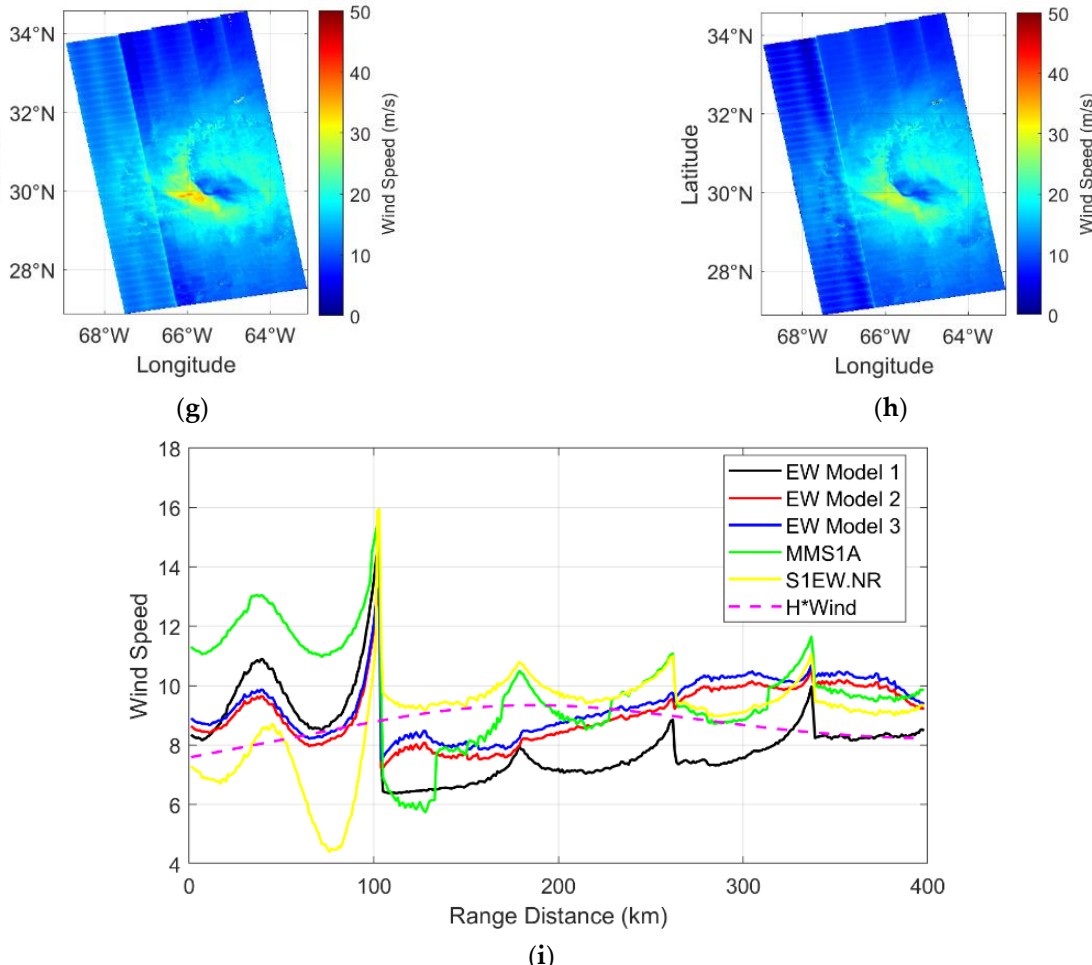

**Figure 12.** A case study of Tropical Storm Karl for model comparison. Sentinel-1 imageries in (**a**) VV-polarization; (**b**) VH-polarization. (**c**) The collocated H*Wind data. Wind retrievals of (**d**) EW Model 1; (**e**) EW Model 2; (**f**) EW Model 3; (**g**) MMS1A; (**h**) S1EW.NR. (**i**) Averaged retrieval variations in azimuth direction according to the area framed in red in (**d**).

In the area framed in red in Figure 12d, wind variations were averaged in the azimuth direction and then compared in Figure 12i. A smaller rise and fall in the winds retrieved by EW Model 2 and 3 demonstrate that the impact of thermal noise was partially weakened. Even though the EW Model 1 was developed with the same MLR method as Models 2 and 3, its retrievals had a larger variation. For all the points whose wind speeds were lower than 10 m/s, the RMSE values were 2.46, 2.06, 2.30, 3.52, and 2.52 m/s for EW Model 1, 2, 3, MMS1A, and S1EW.NR, respectively. These results show the benefit of the VV and VH combination.

## 5. Discussion

As shown in Section 4.1, on average, our SAR retrievals were lower than SMAP measurements if the wind exceeded about 30 m/s. When the wind increased, the difference became larger. To investigate this reason, we compared the relationships between SAR VH-polarized NRCS and the winds measured using SFMR and SMAP. As shown in Figure 4 and Table A1, among the parameters of NRCS, the incident angle, and wind direction, VH NRCS had the strongest impact on wind speed retrieval. Thus, we used it as a medium to compare two different wind measurements and showed the variation characteristics of retrieval in different wind regimes.

The results are shown in Figure 13. Black solid and dashed curves represent the mean variations of scatters with different colors. According to the two curves, the difference in

SAR retrievals and SMAP winds had two main reasons. First, if the wind exceeded 30 m/s for the same VH NRCS, the SMAP measurement would be higher than SFMR on average. Second, the difference between SFMR and SMAP increased with increasing VH NRCS. As a result, when we used the model developed from SFMR data to retrieve wind, the results were close to SFMR and lower than SMAP data. This difference became larger with increasing wind.

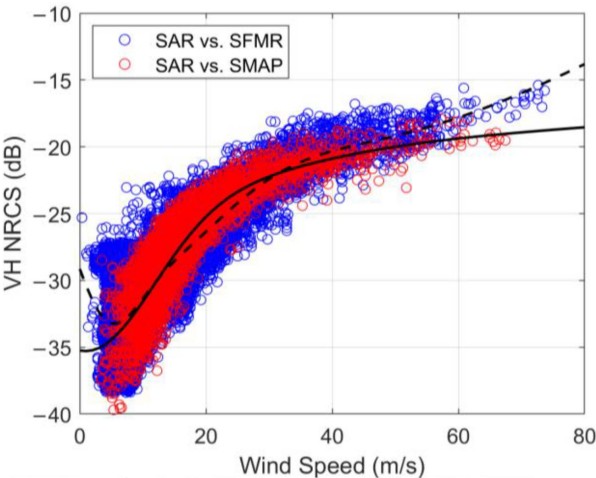

**Figure 13.** Relationships between Sentinel-1 VH NRCS and wind speeds measured using SFMR and SMAP according to our samples. The black dashed curve stands for the mean variation in SAR and SFMR. The black solid curve stands for the mean variation in SAR and SMAP.

In the proposed models, wind direction is one of the inputs in EW and IW Model 3. However, according to the MLR results, it may not be a key parameter in wind speed retrieval if VV- and VH-polarized imageries are combined. As shown in Table A1, most $A_4$ and $A_{i,4}$ are at least one or two magnitudes smaller than other coefficients. It means that wind speed and direction are generally not related or are weakly related to each other. This is true for randomly selected sea areas in nature. Wind direction just modulates the relationship between co-polarized backscatter and wind speed by influencing the local incident angle in a cosine way, which is hard to fit using MLR. As a result, although Model 3 has a wind direction coefficient, it is not much better than Model 2 (see Table 2), indicating that the influence process of wind direction on wind retrieval is not essential if VV- and VH-polarized imageries are combined, while Model 2 and 3 are an evident improvement with respect to Model 1, indicating that the dual-polarization combination is an effective method to improve retrieval accuracy.

Under TC conditions, extreme wind speeds are always accompanied by intense rainfall, which can generate an error for wind retrieval. Since the SFMR instrument can measure contemporaneous wind speed and rain rate along the flight track, its data are an excellent reference to evaluate the impact of rainfall on wind retrieval [39]. In this study, the wind retrieval bias, which was equal to the proposed model retrieval minus the SFMR wind speed, was compared with the SFMR rain rate and wind speed. These results were used to measure the impact of rainfall on wind retrieval. For three models of each mode, their bias distributions were similar. Thus, biases were averaged from the results of Models 1, 2, and 3. As shown in Figure 14, although high rain rates are generally accompanied by high winds, there is no evidence that the rainfall had a definite impact on wind retrieval. For all points, the bias was around zero on average. Due to this ambiguous pattern, reducing errors with rain rate or further retrieving rain rate is still difficult.

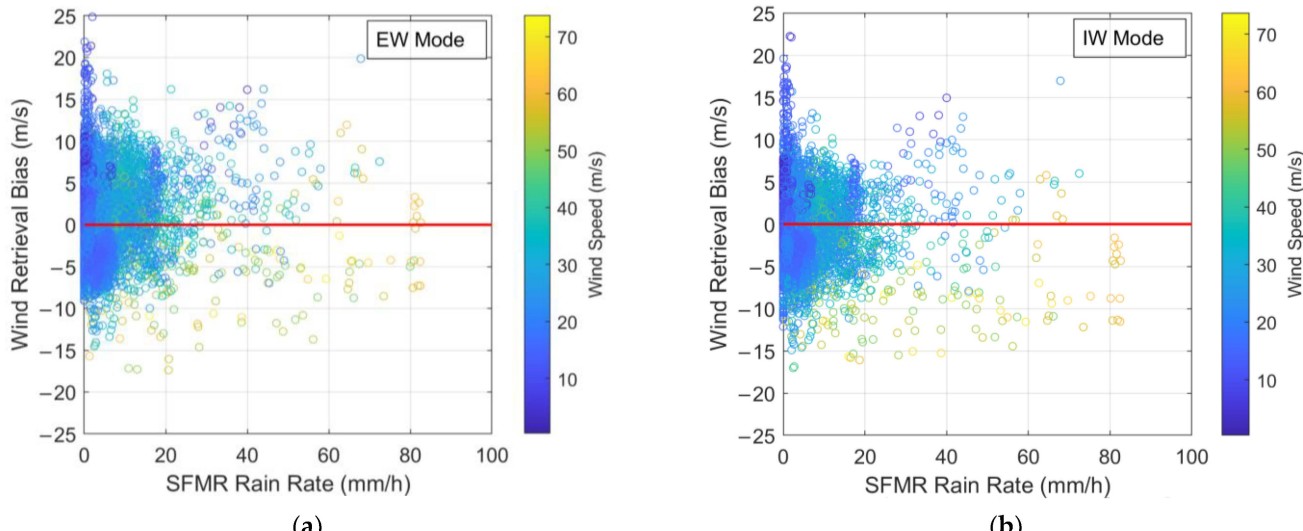

**Figure 14.** Comparisons between wind retrieval bias, SFMR rain rate and wind speed for the proposed models of (**a**) EW mode; (**b**) IW mode. For each mode, results are averaged from Model 1, 2, and 3.

## 6. Conclusions

Inspired by the CWAVE wave retrieval models, this study proposes a new approach for SAR ocean surface wind speed retrieval in a grid spacing of 1 km with the MLR method. To develop and validate new retrieval models, Sentinel-1 dual-polarized (VV + VH) observations were acquired from 28 TC cases over different ocean basins. Collocated wind data were collected from multiple sources, such as SFMR, SMAP, dropsonde, ECMWF ERA5, and H*Wind.

According to the comparison between NRCS, incident angle, and wind speed, VV- and VH-polarized imagery can distinguish wind speeds well, respectively, within low wind regions and moderate-to-high wind regions. Based on these advantages, new models were designed with the combination of VV- and VH-polarized signals. In addition, the GMF form was changed from the traditional $\sigma_0$ output to the new $U_{10}$ output, making the retrieval straighter and faster. Three schemes were investigated for inputs, named after Model 1 (inputs: $\sigma_0^{VH}$, $\theta$), Model 2 (inputs: $\sigma_0^{VH}$, $\theta$, $\sigma_0^{VV}$) and Model 3 (inputs: $\sigma_0^{VH}$, $\theta$, $\sigma_0^{VV}$, $\varphi$). According to the MLR results, the model with more inputs could fit data better. However, the improvement from Model 2 to Model 3 was not as significant as from Model 1 to Model 2.

Validation shows better consistencies between the results of proposed models and SMAP and dropsonde winds compared with traditional VH-polarization GMFs in the piecewise function. The big difference between SAR retrievals and SMAP winds over 30 m/s was mainly caused by the observational difference in SFMR and SMAP. A retrieval case study on Hurricane Michael indicated that our models could provide the wind information of a fine structure, for example, TC eye, eyewall, and rainbands. A case study of Tropical Storm Karl suggested that the impact of thermal noise on wind retrieval could be partially reduced using Models 2 and 3. These results indicate that the use of MLR and the combination of VV- and VH-polarized imageries could improve wind retrieval accuracy. In addition, the input of wind direction is not essential if the combination has been utilized. Under TC conditions, although intense rainfall is often accompanied by high winds, our study showed no evidence that rainfall had a definite impact on wind retrieval. Even with rainfall, the retrieval bias was around zero on average. As a result, it is still difficult to reduce errors with rain rate or further retrieve rain rate.

**Author Contributions:** Initiation of the idea: Y.G.; Data processing and model proposing: Y.G.; Writing and editing: all authors contributed; Supervision: Y.W.; Funding acquisition: Y.G. All authors have read and agreed to the published version of the manuscript.

**Funding:** This study is supported by the Natural Science Foundation of Shandong (Grant ZR2022QF069).

**Data Availability Statement:** The data are available at https://pan.baidu.com/s/19A7SVd5fuQscLn2 DiVruwg (accessed on 30 August 2023).

**Acknowledgments:** We thank the European Space Agency for making the Sentinel-1 data publicly available. The authors thank the NOAA/Atlantic Oceanographic and Meteorological Laboratory/Hurricane Research Division (https://www.aoml.noaa.gov/ (accessed on 20 February 2023)) for providing the SFMR and Dropsonde data and the Remote Sensing Systems (RSS, https://www.remss.com/ (accessed on 21 March 2022)) for providing SMAP data. We thank the European Centre for Medium-Range Weather Forecasts for ERA5 data and Risk Management Solutions for H*Wind data. We thank the National Hurricane Center and the Joint Typhoon Warning Centre for TC best tracks.

**Conflicts of Interest:** The authors declare no conflict of interest.

## Appendix A

In this section, the coefficients are listed in Table A1 for the proposed wind retrieval models, i.e., EW Model 1, 2, 3 and IW Model 1, 2, 3. The coefficients in their correction functions are listed in Table A2.

**Table A1.** Values of all coefficients in the proposed wind retrieval models.

| Model | Coefficient | | | | |
|---|---|---|---|---|---|
| EW Model 1 | $A_0$ 134.948527 | $A_1$ 8.535906 | $A_2$ 1.1293905 | $A_{1,1}$ 0.1422056 | $A_{1,2}$ 0.038811 |
| | $A_{2,2}$ 0.003917 | | | | |
| EW Model 2 | $A_0$ 143.812413 | $A_1$ 11.067208 | $A_2$ 2.355905 | $A_3$ $-0.307838$ | $A_{1,1}$ 0.204342 |
| | $A_{1,2}$ 0.036087 | $A_{1,3}$ $-0.071111$ | $A_{2,2}$ $-0.023669$ | $A_{2,3}$ $-0.064649$ | $A_{3,3}$ $-0.035267$ |
| EW Model 3 | $A_0$ 147.348198 | $A_1$ 11.398898 | $A_2$ 2.377266 | $A_3$ $-0.440641$ | $A_4$ 0.000234 |
| | $A_{1,1}$ 0.209036 | $A_{1,2}$ 0.035286 | $A_{1,3}$ $-0.076520$ | $A_{1,4}$ $-0.000547$ | $A_{2,2}$ $-0.023973$ |
| | $A_{2,3}$ $-0.065019$ | $A_{2,4}$ $-0.000177$ | $A_{3,3}$ $-0.033961$ | $A_{3,4}$ 0.000105 | $A_{4,4}$ $-0.000017$ |
| IW Model 1 | $A_0$ 185.593357 | $A_1$ 12.465933 | $A_2$ 1.315279 | $A_{1,1}$ 0.141039 | $A_{1,2}$ $-0.054268$ |
| | $A_{2,2}$ $-0.029085$ | | | | |
| IW Model 2 | $A_0$ 203.549220 | $A_1$ 15.088689 | $A_2$ 1.653653 | $A_3$ $-0.714153$ | $A_{1,1}$ 0.249729 |
| | $A_{1,2}$ $-0.015968$ | $A_{1,3}$ $-0.085755$ | $A_{2,2}$ $-0.027735$ | $A_{2,3}$ $-0.050190$ | $A_{3,3}$ $-0.034910$ |
| IW Model 3 | $A_0$ 217.780636 | $A_1$ 16.327531 | $A_2$ 2.159972 | $A_3$ $-1.552834$ | $A_4$ $-0.163730$ |
| | $A_{1,1}$ 0.269266 | $A_{1,2}$ $-0.016449$ | $A_{1,3}$ $-0.108816$ | $A_{1,4}$ $-0.003335$ | $A_{2,2}$ $-0.035309$ |
| | $A_{2,3}$ $-0.041120$ | $A_{2,4}$ 0.000859 | $A_{3,3}$ $-0.020604$ | $A_{3,4}$ 0.001688 | $A_{4,4}$ 0.000183 |

**Table A2.** Values of all coefficients in the correction functions of the proposed wind retrieval models.

| Correction Coefficient | EW | | | IW | | |
|:---:|:---:|:---:|:---:|:---:|:---:|:---:|
| | **Model 1** | **Model 2** | **Model 3** | **Model 1** | **Model 2** | **Model 3** |
| *a* | 0.73 | 0.74 | 0.74 | 0.70 | 0.72 | 0.74 |
| *b* | 1.12 | 1.11 | 1.11 | 1.13 | 1.12 | 1.11 |

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
