# Peer review of "A New Approach for Ocean Surface Wind Speed Retrieval Using Sentinel-1 Dual-Polarized Imagery"

_remotesensing, doi:10.3390/rs15174267_

Round 1

Reviewer 1 Report

In figures 5 and 6, in addition to the line of the corresponding main diagonal, add a line of linear regression. The same remark to figures 8-9.

In Figures 5 and 6, instead of markers, draw the data plane as in Figures 8-9.

The distribution plane in Figure 7 is best shown as histograms.

Table 3. RMSE is useful when a linear relationship is observed. According to Figure 8, there is no such dependence. You need to justify the use of RMSE or apply another comparison criterion.

Same remarks for Table 4 and Figure 9.

Author Response

The paper has been revised according to your comments and suggestions. Please see the attachment.

Reviewer 2 Report

The authors develop a new method for wind speed retrieval with the combination of SAR dual-polarized signals.The results suggest that the use of MLR and the dual-polarization combination can improve 21

SAR wind retrieval accuracy. It is overall a good piece of work to be publshed, though major modifications still need to be addressed during the revisions.

1. In the abstract, the abbreviation of VV-polarized and VH-polarized should be defined when they were first used.

2.Around lines 236 to 237, I suggest the calculation of Cor and RMSE to be given in the content.

3.Line 370, the data access is much better to be the direct link, instead of the author.

The English is good.

Author Response

(The authors gave the same response as above.)

Reviewer 3 Report

A New Approach for Ocean Surface Wind Speed Retrieval Using Sentinel-1 Dual-Polarized Imagery

Manuscript submitted to Remote Sensing

General overview

The paper has an innovative content about wind estimations based on satellite data. The use of a new method considering the dual-polarized sensing mode reveals a powerful manner to obtain wind speed and direction with less uncertainty.

The structure follows the standards, the text is clear and well-written and the conclusions are fully supported by the obtained results. I have minor comments to improve the manuscript. My opinion is accepted with minor revisions, almost “accepted as it is”.

Specific comments

Line 78: performs instead of “preforms"

Line 86: SMAP: I think it is better to present the full name before to introduce the acronym, which is used sometimes in the text

Lines 176-178: phrases should be merged: “To note, a SAR image of Hurricane Michael observed on 23:44 UTC 9 October 2018 could be matched both with SFMR and SMAP (Figure 2), where the storm eye is obvious.”

Lines 194-195: phrases should be merged: “After data collocation, the distribution of NRCS was compared with incident angle and wind speed (Figure 4).”

Line 325-327: phrases should be merged. “Since the SFMR instrument can measure contemporaneous wind speed and rain rate along flight track, its data is an excellent reference to evaluate the impact of rainfall on wind retrieval [37].”

Not applicable.

Author Response

(The authors gave the same response as above.)

Reviewer 4 Report

The authors proposed a new approach to retrieve sea surface wind speed from Sentinel-1 dual-polarized SAR imagery. The paper is readable and interesting. With the new models, one can get TC surface wind distribution by inputting VV NRCS, VH NRCS, incident angle and wind direction. The new models are easier to use than traditional GMFs. Validation demonstrated that the proposed models are more accurate than traditional GMFs. The results are convincing. In addition, I think the proposed method can not only be used for Sentinel-1 SAR but also for other SAR instruments. The paper is well organized and written. The following are my minor suggestions.

(1) As mentioned in Line 91, the proposed model used the same fit method as CWAVE, which is a wave retrieval model family. So, more new papers about wave retrieval should be cited in ‘1. Introduction’ to show the state of art, not only the CWAVE family.

(2) Line 106-112. The author introduced the different parameters between Sentinl-1 EW and IW modes, but those are not enough. Please add image numbers for two modes to make dataset clear.

(3) Figure 4. Please mark ‘IW Mode’ and ‘EW Mode’ in figures as others.

(4) Figure 4 (b) and (d). NESZ is the noise floor. Why there are some points lower than NESZ?

(5) In ‘3. Model Development’ Equation (1). All three schemes have VH NRCS input, but Model 1 doesn’t have VV NRCS. I see the Model 1 can still retrieve winds. Does that mean the VV NRCS is less important than VH NRCS?

(6) Figure 10. There are few dropsonde measurements. Can author collect more?

(7) Figure 12 (c). The image missed a corner, compared with other retrieval maps. Please correct.

(8) In ‘4. Validation and Comparison’, there are not comparison between SAR retrievals and ERA5. Why did ERA5 data only be used for providing wind directions?

(9) In ‘6. Conclusion’, author could mention that the new model is suitable for wind retrieval in 1-km grid spacing.

(10) Line 436-438. Please organize ‘Author Contributions’ in one paragraph.

Author Response

(The authors gave the same response as above.)

Round 2

Reviewer 2 Report

The authors have addressed all my comments. I recommend this manuscript to be published.